# Extreme Tolerance of Extraocular Muscles to Diseases and Aging: Why and How?

**DOI:** 10.3390/ijms25094985

**Published:** 2024-05-03

**Authors:** Angelina Titova, Sergey Nikolaev, Airat Bilyalov, Nikita Filatov, Sergei Brovkin, Dmitrii Shestakov, Igor Khatkov, Ekaterina Pismennaya, Vyacheslav Bondarev, Margarita Antyuxina, Elena Shagimardanova, Natalia Bodunova, Oleg Gusev

**Affiliations:** 1Institute of Fundamental Medicine and Biology, Kazan Federal University, 420008 Kazan, Russia; 2SBHI Moscow Clinical Scientific Center Named after Loginov MHD, 111123 Moscow, Russia; 3Ministry of Health Kursk Region, 305000 Kursk, Russia; 4Regional Pathology Bureau, 305007 Kursk, Russia; 5Regional Perinatal Center, 305007 Kursk, Russia; 6Intractable Disease Research Center, Graduate School of Medicine, Juntendo University, Tokyo 113-8421, Japan; 7Life Improvement by Future Technologies (LIFT) Center, 121205 Moscow, Russia

**Keywords:** extraocular muscles, muscular dystrophies, muscular aging

## Abstract

The extraocular muscles (EOMs) possess unique characteristics that set them apart from other skeletal muscles. These muscles, responsible for eye movements, exhibit remarkable resistance to various muscular dystrophies and aging, presenting a significant contrast to the vulnerability of skeletal muscles to these conditions. In this review, we delve into the cellular and molecular underpinnings of the distinct properties of EOMs. We explore their structural complexity, highlighting differences in fiber types, innervation patterns, and developmental origins. Notably, EOM fibers express a diverse array of myosin heavy-chain isoforms, retaining embryonic forms into adulthood. Moreover, their motor innervation is characterized by a high ratio of nerve fibers to muscle fibers and the presence of unique neuromuscular junctions. These features contribute to the specialized functions of EOMs, including rapid and precise eye movements. Understanding the mechanisms behind the resilience of EOMs to disease and aging may offer insights into potential therapeutic strategies for treating muscular dystrophies and myopathies affecting other skeletal muscles.

## 1. Introduction

The extraocular muscles (EOMs) are unique in several aspects: They represent the fastest and most fatigue-resistant muscles within the human body [1,2,3]. Extraocular muscles (EOMs) predominantly exhibit impairment in conditions such as myasthenia gravis and mitochondrial myopathies, yet, remarkably, they are spared from various muscular dystrophies, including Duchenne, Becker, limb-girdle, and congenital muscular dystrophies. Furthermore, EOMs demonstrate particular resistance to amyotrophic lateral sclerosis (ALS) [4,5].

The complexity of the actions performed by the extraocular muscles (EOMs) is reflected in their anatomical and physiological characteristics. Morphologically and in terms of their molecular composition, they significantly differ from the muscle fibers (MFs) of other skeletal muscles. The gene expression profile of the EOMs is distinct from that of limb muscles, with differences encompassing over 330 genes involved in metabolic pathways, structural components, development markers, and regenerative processes. Unlike skeletal muscles, the EOMs predominantly utilize an aerobic pathway for carbohydrate metabolism and relies directly on the glucose influx from the blood. This metabolic strategy enables them to be among the fastest muscles in the body while also being exceptionally resistant to fatigue [6].

Our review focuses on understanding the cellular and molecular basis of the unique properties of extraocular EOMs, which enable them to resist muscular dystrophies and aging. Comprehending how EOMs counteract diseases that lead to degeneration and functional loss in skeletal muscles could lay the groundwork for developing gene and cell therapies and medical devices for treating dystrophies and myopathies of various etiologies.

### 1.1. Structural Characteristics of the EOMs

EOMs enable the eyeball to perform slow pursuit movements and rapid, instantaneous shifts from one fixation point to another. The EOMs comprise six muscles responsible for controlling eyeball movement and one muscle, the levator palpebrae, which governs eyelid motion: superior/inferior/lateral/medial rectus and superior/inferior oblique (Figure 1). Unlike skeletal muscles, the EOMs are innervated by cranial nerves rather than motor neurons located in the spinal cord [7]. 

The development of EOMs originates from the prechordal and cranial paraxial mesoderm [8]. The mesodermal mesenchyme that forms the EOMs is interspersed with cells of the cranial neural crest, which contribute to the formation of the sclera and the connective tissue surrounding the muscles [9,10,11,12,13,14,15,16]. Experiments involving the ablation of the neural crest suggest an interactive process in the formation of the EOMs between the mesoderm and the neural crest. However, due to variations in experimental models, the involvement of the neural crest in the development and organization of the EOMs remains somewhat controversial. Nonetheless, given the contribution of cranial neural crest cells to the development of craniofacial bones, connective tissue, nerves, and the eye, it is plausible that they play a significant role in the organization of the EOMs [9,17,18,19,20,21,22]. 

During embryogenesis, EOMs undergo the same stages of myogenesis as skeletal muscles. However, unlike skeletal muscles, EOMs retain “immature” protein isoforms, including embryonic myosin [23]. It has also been demonstrated that only minor changes occur with age in EOMs, and they are less susceptible to aging compared to skeletal muscles [24,25]. This persistence of “immature” myosins may facilitate the continuous self-renewal and regeneration of EOMs, potentially accounting for their particular resistance to muscular dystrophies and aging.

### 1.2. Muscle Fibers of Extraocular Muscles

Skeletal muscle fibers (MFs) are conventionally classified as slow (type I) and fast (type II with subtypes a, b, and x) to reflect their contractile and metabolic properties [6]. However, this classification is not applicable to the EOMs [26,27]. In contrast to skeletal muscles, they possess a distinct classification scheme based on their contractile abilities, anatomical location, and innervation [28].

Each EOM is subdivided into at least two primary layers: the “orbital” layer, featuring thinner MFs adjacent to the bony walls of the orbit (forming the central fascicle in the oblique muscles), and the internal “global” layer, containing thicker MFs directly abutting the eyeball and optic nerve (Figure 2) [29,30]. In these layers, the muscle fibers can be either singly or multiply innervated MFs [29,30]. The levator palpebrae superioris muscle, uniquely among all the EOMs, is not multilayered and comprises solely singly innervated MFs [31].

During development, skeletal muscles express both isoforms of myosin heavy chains (MyHCs) involved in development: MyHCemb and MyHCneonatal. However, post-development, only the expression of a single adult isoform of MyHC I, IIa, IIb, and IIx is retained. It is the type of expressed MyHC isoform that determines the contraction speed, strength, and ATP consumption by the muscles [32]. 

The expression of embryonic and neonatal MyHC isoforms is retained in mature EOMs, which also express nearly all known MyHC isoforms (Table 1). Additionally, the expression of more than one MyHC isoform in individual muscle fibers has been documented [33,34,35,36,37]. Beyond embryonic forms, two ancient MyHC isoforms, MYH14/7b and MYH15, have been identified in adult EOMs [38,39,40].

Nevertheless, the presence of all MyHC isoforms has not always been confirmed by researchers studying EOMs. Occasionally, they were unable to identify certain isoforms, referring to such MHCs in their studies as unidentified isoforms [41,42,43,44,45,46].

In 1971, Mayr proposed a classification of EOM fiber types and divided them into six groups: two groups in the “orbital” layer and four in the “global” layer, which remains the standard classification to this day (Table 1: Types of EOM muscle fibers) [28,47].

Singly innervated, fast-twitch, and fatigue-resistant in the “orbital” layer.Multiply innervated with both fast-twitch and slow-twitch fibers, exhibiting variable fatigue resistance in the “orbital” layer.“Red,” singly innervated, fast-twitch, and fatigue-resistant in the “global” layer.“White,” singly innervated, fast-twitch, and exhibiting low fatigue susceptibility in the “global” layer.“Intermediate,” singly innervated, fast-twitch with moderate fatigue resistance in the “global” layer.Multiply innervated, slow-twitch, and fatigue-resistant in the “global” layer.

Despite the established standard, the classification of EOM fiber types in the literature is quite heterogeneous. Some studies describe only five fiber types of EOMs in mammals: four in the “global” layer and three in the “orbital” layer, with two identical EOM types present in both layers [48]. In a subsequent study, the same researchers identified six distinct types, with four identical types in each layer [49]. Other studies also classified six different types, but with three distinct EOM types in each layer [30].

Some studies indicate that EOMs are structurally more complex compared to those of other mammals. In 2000, a group of researchers, in their efforts to classify the neuromuscular MFs of EOMs, identified a third layer known as the marginal zone. Within this zone, they described three types of MFs: singly innervated muscle fibers and two types of multiply innervated muscle fibers [50]. However, in later papers by other researchers, there is no mention of a third layer in the human EOMs.

Despite extensive research on EOMs, scientists have yet to reach a consensus on the distribution and quantity of muscle fiber (MF) types in the EOMs. Disagreements among researchers may stem from varying methodologies employed in their studies, the presence of MFs expressing multiple types of MyHCs, and the unique and complex structural organization of the EOMs. The lack of consistency in classifying EOM MFs further underscores their uniqueness. In addition to the multitude of MyHC types found in the EOMs, which are not present in other skeletal muscles, these muscles also exhibit distinct motor innervation patterns.

### 1.3. Motor Innervation

The ratio of nerve fibers to muscle fibers in the EOMs is more than tenfold higher (ranging from 1:3 to 1:5 or 1:12) compared to skeletal muscles (ranging from 1:50 to 1:125) [51,52,53]. The architecture of motor endplates depends on the structure, physiology, and histochemistry of the MFs [54,55,56,57]. Muscles are categorized into two primary groups based on their contraction capabilities: phasic (twitch) and tonic muscle fiber [51,52,53]. Tonic muscle fibers maintain static tension, or tone. In contrast, phasic muscle fibers can perform rapid contractions but are incapable of sustaining the achieved level of shortening over extended periods [58,59,60].

Muscle fibers characterized by tonic contraction types are prevalent in amphibians and reptiles and have also been identified in the mammalian EOMs [61,62,63,64,65,66]. Across all the studied species, these tonic muscle fibers exhibit multiple innervations [66,67,68,69,70]. In contrast, phasic (twitch) muscle fibers possess a single innervation and are broadly classified into two subclasses within skeletal muscles based on their contraction velocity: slow (type I) and fast (type II) [71]. 

The evolutionarily tonic type of contraction represents an ancient form, as evidenced by the process of muscle development and during embryogenesis, where MFs exhibit multiple innervations. During development, motor neurons extend their axons to the prospective sites of skeletal muscle formation, coinciding with the emergence of muscle tubes. Multiple axons grow towards the initial contact site, where multiply innervated muscle fibers occur at this juncture. However, in skeletal muscles, only the most active axon forms the definitive neuromuscular synapse [58].

Axons innervating phasic (twitch) muscle fibers terminate in a single neuromuscular junction (NMJ), referred to in the literature as an en plaque ending. Conversely, tonic muscle fibers (MyHCsto/I) in the EOMs receive multiple innervations, with grape-like NMJ clusters, which are referred to in the literature as en grappe endings (Figure 3).

Recent investigations in EOMs containing MyHCeom have revealed a novel type of NMJ, characterized by multiterminal endings distributed along the entire length of the MFs. Contrary to the en grappe NMJ, which is traditionally aligned in a row on one side of the MFs, this multiterminal NMJ is situated on both sides of the MFs. Furthermore, the study demonstrated that en grappe endings were exclusively identified in myofibrils harboring MyHCsto/I [72].

The distribution of mitochondria within EOM fibers varies based on the type of innervation and the specific EOMs. Research from the 1970s to the 1990s documented diverse mitochondrial compositions in EOM fibers, associated with their contraction patterns. Khairy et al. noted that fibers in the “orbital” layer exhibit a relatively higher mitochondrial content compared to those in the “global” layer [73]. Demer et al. suggested that the differences in the mitochondrial content between the “orbital” and “global” layers could be attributed to their functional specialization [74]. Other researchers have indicated that EOMs, in general, possess a higher mitochondrial density than skeletal muscles [6].

Singly innervated MFs exhibit phasic (twitch) contraction types and contain a high density of mitochondria throughout their length (Figure 4) [75]. In the “global” layer, red, singly innervated MFs possess a greater mitochondrial content, whereas pale singly innervated ones have fewer [76]. Conversely, the multi-innervated MFs of the “global” layer display a tonic contraction type and house only a scant number of mitochondria. Meanwhile, “orbital” MFs can demonstrate phasic (twitch) contractions and an NMJ resembling en plaque formations in the muscle’s midsection coinciding with en grappe terminations and tonic contraction types in the MFs peripheral regions [77]. Consequently, the central portion of the “orbital” multi-innervated MFs is also rich in mitochondria, similar to the singly innervated fibers [76].

Based on the type of innervation, three categories of MFs can be distinguished [30]: Singly innervated (twitch) muscle fibers, corresponding to fast skeletal MFs. This type constitutes the dominant population in both the “global” (90%) and “orbital” (80%) layers; electrical stimulation of the innervating axon triggers a twitch response based on the “all-or-nothing” principle.Multiply innervated (tonic) MFs in the “global” layer (10%) with NMJs en grappe [78]. Electrical stimulation of the innervating axons causes graded tonic contractions [79].Multiply innervated MFs in the “orbital” layer (20%) with NMJs en grappe and en plaque, exhibiting corresponding dynamics of contraction in the central and distal parts upon electrical stimulation [80].

In addition to the presence of multiply innervated MFs, another distinctive feature of EOMs is the preservation of expression in adult muscles of the gamma-subunit of the fetal acetylcholine receptor (γ-AChR), which, in skeletal muscles, occurs only during the period of intrauterine development or in denervated MFs, and within the first two weeks after birth, it is replaced by the ε-subunit (γ/ε switching) [72,81,82]. 

In EOMs, both γ-AChR and ε-AChR subunits are preserved [72,83]. The fetal γ-subunit is essential for the proper maturation of the neuromuscular synapse [84,85,86]. Mutations in γ-AChR can lead to both lethal and non-lethal alterations, such as multiple pterygium syndrome (also known as Escobar syndrome, a form of congenital myasthenic syndrome) [87,88,89].

The particular composition of AChR subunits in EOMs is likely related to the type of MFs and their physiological properties. AChRs containing the γ-AChR subunit differ from those containing the ε-AChR subunit in terms of their ionic conductivity and ion channel opening rates [90]. The ion channel opening time during limb muscle development decreases from 11 ms to 6 ms when switching from the fetal γ-subunit to the adult ε-subunit of AChR [90]. Interestingly, fast MFs expressing MyHCIIa exclusively contain ε-AChR subunits, whereas tonic and slow MyHCsto/I with the NMJ en grappe express γ-AChR subunits [72]. The difference in the AChR subunit expression corresponds to the difference in the contraction speed between these two types of MFs. 

During evolution, EOMs have retained the tonic contraction type, which seems necessary for performing specialized eye movements, such as tracking or sustained fixation on an object. However, maintaining tonic contraction in MFs requires multiple innervations not found in mature skeletal muscles. In addition to having a tonic, slow contraction in specific MFs, EOMs are also the fastest muscles in the body. These functional characteristics of EOMs are manifested in the presence of MyHC types not found in skeletal muscles, the retention of the fetal γ-AChR subunit, and differences in the cytoskeletal structure and surrounding connective tissue of the MFs.

### 1.4. Cytoskeleton and Basal Membrane of EOMs

Another factor contributing to the disease resistance in EOMs might be attributed to the architectural uniqueness of their cytoskeleton. The muscle cytoskeleton comprises a complex network of proteins, organized to interlink and anchor cellular structural components such as myofibrils, nuclei, and sarcolemma, regulating cellular morphology and facilitating the transduction of myofibrillar movements into the surrounding tissues. 

Desmin is the first muscle-specific protein discovered during muscle development, and it is such a fundamental component of muscle fibers that it is used as a marker for muscle tissue. Desmin is considered essential for maintaining the structural and functional integrity of the postsynaptic apparatus [91,92].

Although muscle fibers can develop in the absence of desmin, defects in the genes encoding this protein result in severe myopathy [93]. Nevertheless, it has been discovered that a portion of healthy, undamaged EOM fibers in humans lacks desmin, constituting a true paradigm shift in the understanding that desmin is essential for all muscle fibers. Furthermore, despite the absence of desmin, EOMs remain unaffected in desmin-related myopathy [94].

The study conducted by Jing-Xia Liu and Fatima Pedrosa Domellöf confirmed the absence of desmin in many motor endplates of human EOMs and further revealed a lack of desmin in the immediate vicinity of the NMJ, in contrast to the abundant presence of desmin at the NMJ observed in limb muscles. A concise summary of their findings is presented in Figure 5 [94].

These findings suggest that desmin does not play the same role in EOMs as it does in limb muscles. Despite the absence of desmin, these neuromuscular synapses exhibit normal morphology and the normal distribution of many crucial synaptic molecules, including AChR subunits; neurofilaments and synaptophysin; laminin isoforms α2, α4, α5, and β2; and various isoforms of Wnt, neurotrophins, and their receptors in healthy human EOMs [95,96,97,98,99]. Consequently, the lack of desmin in neuromuscular synapses is not considered a defect in human EOMs. 

In the context of these studies, it is particularly noteworthy that mice with desmin deficiency exhibited a shift in MF types towards slow, fatigue-resistant fibers [100,101]. Additionally, one of the functional characteristics of EOMs is their high resistance to fatigue.

Another factor contributing to resilience and a significant distinction from skeletal muscle is the composition of the basal membrane, which protects EOMs from congenital muscular dystrophy with Lna2 deficiency. The basal membrane is situated on the immediate surface of skeletal muscle fibers, with laminins serving as one of the principal components of the basal membrane. Laminins are associated with receptors that span the thickness of the cell membrane, thereby attaching to the cytoskeleton within the muscle fiber. It has been demonstrated that the basal membrane surrounding the EOM fibers in humans contains not only the typical α-chain laminin isoform present in adult skeletal muscle fibers but also other isoforms that are present in skeletal muscles only during development. Thus, EOMs remain unaffected by the congenital Lna2 defect because they typically possess additional isoforms of the same laminin chain [102]. Additionally, EOMs express the Lutheran protein, which is an α5-chain-specific receptor, not found in limb muscles [102]. The presence of laminin isoforms, absent in adult skeletal muscles, is a common feature of all mature EOMs, and this may indeed serve as an additional protective mechanism against diseases. In one study, it was shown that the expression of laminin (Lama2) in EOM fibroblasts was increased 17-fold compared to skeletal muscle fibroblasts, consistent with the observation that EOMs have much denser connective tissue compared to skeletal muscles overall, including leg muscles [102].

### 1.5. Connective Tissue

Typically, each MF in the EOMs is surrounded by connective tissue and is not in direct contact with adjacent fibers, as is commonly observed in other skeletal muscles. This confers a round contour to the EOM fibers, whereas skeletal muscles exhibit a polygonal shape due to the close proximity of neighboring MFs [103].

The connective tissue of the EOMs contains fibroblasts that exhibit properties distinct from those in other skeletal muscles, potentially contributing to the unique response of the EOMs and their involvement in inflammatory diseases such as Graves’ ophthalmopathy (thyroid eye disease) [104,105,106]. Another feature is the distinct genomic profile of EOM fibroblasts compared to those of skeletal muscles. Orbital fibroblasts develop from neural ectoderm, unlike fibroblasts from other regions, which are of mesenchymal origin [107,108]. Consequently, significant differences in the transcription factors of EOMs can be anticipated [108].

Genes associated with a differential immune response have been identified. Specifically, in EOM fibroblasts, there was an increased expression of CD59a, a complement system regulator that impacts the severity of myasthenia [109,110,111] and IL-6, which is involved in Graves’ ophthalmopathy [112,113]. The response of EOMs to immune mediators differs from that of other skeletal muscles, with fibroblasts playing a significant role in this response [114,115]. Furthermore, the genomic profile of EOMs demonstrated the elevated expression of calreticulin, which may act as an autoantigen in Graves’ ophthalmopathy [116].

### 1.6. Muscle Fiber Metabolism and Antioxidant Capacity

The enhanced expression of factors such as IL-6 in EOM fibroblasts may increase glucose uptake in MFs [109]. Concurrently, the expression profile of MFs themselves suggests that they utilize aerobic carbohydrate metabolism and are dependent on blood glucose levels as an energy source. Support for this theory is evidenced by the reduced expression of phosphoglucomutase (a key enzyme in glycogen synthesis), phosphorylase kinase (a principal regulator of glycogen breakdown or glycogenolysis), and a decreased glycogen content in EOMs. The deficiency of these enzymes in skeletal muscles leads to type VIII glycogen storage disease and McArdle’s syndrome (type V glycogen storage disease), yet these conditions do not affect the EOMs [6,7]. 

In EOMs, only a minimal amount of glycogen is synthesized, and these muscles do not rely on it as their primary energy source. Instead, they predominantly oxidize glucose transported through the bloodstream to meet their energy demands. Furthermore, the expression profile of EOMs suggests efficient glucose utilization via the Krebs cycle and conversion into energy through oxidative phosphorylation. The reliance of EOMs on oxidative phosphorylation accounts for their sensitivity to mitochondrial myopathies [6].

The energetic metabolism based on blood glucose, together with a high mitochondrial content, suggests a mechanistic basis for the common metabolic adaptations utilized by EOMs to sustain fatigue-resistant, rapid (∼400 Hz) contractions over relatively prolonged periods [117]. It is anticipated that the high mitochondrial content and oxidative phosphorylation would lead to an increase in the levels of reactive oxygen species (ROS) in EOMs, potentially causing significant cellular damage [6]. However, it has been demonstrated that EOMs expresses high levels of mRNA for chemoprotective enzymes, glutathione-S-transferase (GST), and UDP-glucuronosyltransferase (UDP-GT), presumably as an adaptation to the elevated ROS, which would be generated during oxidative phosphorylation in EOMs. This indicates that EOMs have enhanced antioxidant enzyme activity compared to limb muscles [118].

### 1.7. Protection, Regeneration, and Aging of EOMs

Given the resistance of EOMs to multiple myopathies, it is plausible that the progenitor cell niche in EOMs markedly differs from that in other muscle groups. The proportion of satellite cells in undamaged adult EOMs is quintuple that in limb muscles [4,5]. Additionally, activated satellite cells are observed in intact EOMs [119]. A high quantity of progenitor cells is maintained even in the elderly, rendering the EOMs more resistant to age-related degeneration or senescence [4,120]. Progenitor cells in EOMs exhibit an increased capacity for proliferation, differentiation, and self-renewal [4,121]. Furthermore, these cells display the increased expression of trophic factors, including brain-derived neurotrophic factors and nerve growth factors, which may account for their enhanced myogenic activity [122]. These characteristics lead to the hypothesis that EOMs are capable of continuous remodeling throughout life [119].

Continuously activated satellite cells may be associated with a unique and/or more abundant progenitor cell subpopulation in adult EOMs, which enables EOMs to undergo continuous remodeling without stem cell depletion, unlike what is observed in limb skeletal muscles [123]. Similar to limb muscles, EOMs contain Pax7-positive satellite cells, and morphometric analysis of histological sections revealed a significantly higher number of Pax7-positive cells compared to limb skeletal muscles [124,125]. Another progenitor cell population is the EOMCD34 cells, characterized as CD34+/Sca1-/CD31-/CD45-. These EOMCD34 cells are myogenic and are maintained during aging in EOMs [124,125]. In murine models of DMD (Duchenne muscular dystrophy), the EOMCD34 cell population rapidly declines in limb skeletal muscles but is predominantly preserved in EOMs. Interestingly, EOMCD34 cells, derived from EOMs, exhibit increased proliferative capacity compared to their counterparts from limb skeletal muscles [125].

In the extraocular EOMs, there is a complete absence of Pax3-positive myogenic precursor cells, which are typically co-expressed with Pax7 in skeletal muscles and are thought to be responsible for muscle regeneration observed in the absence of Pax7-positive cells [126]. Contrary to the lack of Pax3, it has been demonstrated that there exists a myogenic progenitor cell population expressing the transcription factor Pitx2. Pitx2 is a homeobox transcription factor that plays a pivotal role in myogenesis in the cranial region [127]. Pitx2-positive mononuclear cells are located both in the traditional location of satellite cells and in the interstitial connective tissue of the EOMs. Pitx2 progenitor cells represent another significant population involved in the remodeling and regeneration of the EOMs. Cells expressing Pitx2 are also predominantly preserved in EOMs in murine models of muscular dystrophy and in aging skeletal muscles [128]. Western blotting and immunohistochemistry data showed that 80% of EOMCD34 cells express Pitx2. Knockdown of Pitx2 expression in EOMCD34 cells in vitro reduced the rate of proliferation and disrupted the ability of the cells to fuse into multinucleated muscle tubes [125]. Elevated levels of Pitx2 were maintained in dystrophic and aging EOM cells and EOMCD34 cells compared to limb muscles. Postnatal knockout of Pitx2, specific to skeletal muscles, causes a loss of characteristic expression patterns of myosin heavy-chain (MyHC) isoforms in the EOMs of transgenic mice, including the loss of EOM-specific (MYH13) and alpha-cardiac (MYH6) MyHCs [129,130]. Furthermore, these mice with conditional Pitx2 knockout lose the multiply innervated muscle fibers typically found in the EOMs, making them more phenotypically similar to limb skeletal muscles [130]. These differential requirements for Pitx2 during development and in adulthood may contribute to the constitutive differences between the EOMs and limb skeletal muscles, their preservation in muscular dystrophies, and their resilience to injury and denervation. 

The myogenic environment of the EOMs, enriched with precursor cells, may be sustained by the expression of several neurotrophic factors, which typically diminish in skeletal muscles [131], including insulin-like growth factor-1 and -2 [132,133], brain-derived neurotrophic factor [134,135], glial cell line-derived neurotrophic factor [136,137], and neurotrophin-3. These neurotrophic factors are crucial for the maintenance and development of ocular motor neurons [138,139].

Consequently, a unique environment enriched with neurotrophic factors and progenitor cells renders the EOMs resistant to aging and muscular dystrophies, leading us to the conclusion that the EOMs’ response to damage or the introduction of myotoxic agents may significantly differ from that of skeletal muscles.

### 1.8. Response to Acute Damage by Botulinum Toxin in EOMs

Injections of botulinum toxin into most other skeletal muscles lead to significant MF atrophy [140,141]. However, animal studies have shown that MF atrophy does not occur in the EOMs following botulinum toxin injections [142]. In contrast to the atrophy observed in skeletal muscles, there is evidence describing the hypertrophy of singly innervated MFs in EOMs as a result of botulinum toxin administration [143]. One potential explanation is that the injection of botulinum toxin into the EOMs induces a significant increase in the activation, proliferation, and incorporation of progenitor cells into the existing MFs [144,145].

## 2. EOMs in Diseases

### 2.1. Duchenne Muscular Dystrophy

The cause behind the morphological and functional resilience of EOMs to Duchenne muscular dystrophy and related muscular dystrophies has long been a subject of research inquiry [146,147]. Numerous studies support the hypothesis that the remarkable regenerative capability of EOMs enables them to remain both morphologically and functionally exempt from many forms of muscular dystrophy. As discussed in the preceding section, EOMs not only contain a large population of Pax7-positive myogenic precursor cells but also express an abundant population of Pitx2-positive myogenic precursor cells, which are significantly reduced in other skeletal muscles [148]. 

Hypotheses suggesting why EOMs avoid damage in Duchenne muscular dystrophy converge on the notion that EOM-specific properties may aid in preventing or mitigating the deleterious effects of dystrophin deficiency. As an additional compensatory mechanism beyond regeneration, the hyperexpression of the dystrophin-related protein, utrophin (which can functionally replace dystrophin), has been proposed. Utrophin upregulation has been reported in some, but not all, studies conducted on normal and dystrophin-deficient EOMs. No increase in utrophin was observed in the EOMs of normal dogs or dogs with dystrophin deficiency, nor in the EOMs of control humans or patients [149,150].

Another theory of resistance to Duchenne muscular dystrophy involves the high antioxidant capacity of EOMs [118] and/or the ability to maintain intracellular calcium homeostasis to protect against dystrophin-deficient necrosis. The expression profile of EOMs suggests that they are capable of maintaining intracellular Ca2+ homeostasis through the increased expression of the phospholamban (PLN) gene, which regulates the activity of the sarcoplasmic reticulum Ca2+-ATPase. Decreased PLN levels characterize the abnormal Ca2+ flux observed in cardiac hypertrophy [6].

### 2.2. Amyotrophic Lateral Sclerosis

ALS is a severe neurodegenerative disease that leads to the death of primarily central and peripheral motor neurons. It is characterized by progressive muscle weakness, paresis, and paralysis. Although the progression of the disease can vary significantly, death usually occurs within 3 to 5 years due to respiratory failure. In ALS, all skeletal muscles are affected, and there are virtually no clinical symptoms from the EOMs, even at the late stages of the disease.

Neurons of the oculomotor, trochlear, and abducens nuclei, located in the midbrain and responsible for eye movement, exhibit resistance to degeneration in ALS. This resistance enables patients, even in advanced stages of the disease, to communicate using eye movements, often with the aid of computers [151,152]. Furthermore, there is a vulnerability gradient, where faster motor units are affected earlier than slower ones [153]. Consequently, “fast” muscles, primarily dependent on glycolysis, become paralyzed before slower muscles [154].

Considering the plethora of unique characteristics of EOMs, it is not surprising that in ALS, these muscles remain resistant to degeneration even at later stages of the disease. However, in patients with bulbar onset ALS, impairments in EOM functions, such as saccadic speeds and smooth pursuit movements, have been reported [155]. The primary pathoanatomical findings in the EOMs of such patients included mild hypertrophy, MF atrophy, and increased connective tissue content [23]. Additionally, a reduction in the levels of myosin heavy-chain isoforms MyHCI, MyHCIIa, and MyHCsto in MFs was noted [156].

In a study published in the journal *Investigative Ophthalmology & Visual Science* [40], potential changes in the EOMs responsible for the preservation of their integrity were described. It was demonstrated that the quantity of muscle fibers containing MyHCIIa significantly decreases, while the proportion of MyHCeom increases (in the “global” layer) in patients with ALS [40]. A more pronounced shift towards muscle fibers containing MyHCeom was noted in patients with bulbar onset compared to those with spinal onset. These findings correlate with more pronounced clinical manifestations in the EOMs of patients with bulbar onset [40].

It is noteworthy that no significant differences in the types of MFs containing MyHCIIa or MyHCeom were detected in the “orbital” layer of the EOMs, both in donors with spinal and bulbar onset compared to the control samples. These findings may indicate a higher resilience of the “orbital” layer MF to pathological processes in comparison with the “global” layer [40].

The observed increase in the MF value in the context of an increase in the MF MyHCemay suggests that the neurons responsible for controlling these muscle fibers may be exhibiting increased resistance to ALS [157].

## 3. Conclusions

Highly specialized EOMs fundamentally differ from other muscles, warranting their classification as a distinct allotype of MF. They exhibit distinct responses to damage, aging, and diseases specifically affecting any muscles, remaining anatomically and functionally intact even in late stages of conditions such as Duchenne muscular dystrophy or ALS.

Extraocular muscles (EOMs) not only exhibit resilience to individual pathologies but also differ in their anatomical structure, composition, biochemistry, and physiological function compared to other muscles. EOMs are innervated by cranial nerves rather than spinal motor neurons. Some MFs within EOMs display multiple innervations, forming en grappe endings, a feature not found in any muscle of adult mammals. Additionally, a portion of EOM MFs may simultaneously exhibit both phasic and tonic contraction types, and the MyHC expression pattern encompasses the expression of nearly all known MyHC types.

The preservation of embryonic MyHCs and fetal γ-AChR, the heightened expression of genes associated with growth, development, and regeneration, unique embryology, complex muscle fiber types, diverse innervation patterns, and a metabolism distinct from skeletal muscles all indicate that studying EOMs could aid in the development of potential therapeutic strategies for muscular pathologies linked to diseases, injuries, and aging.

## Figures and Tables

**Figure 1 ijms-25-04985-f001:**
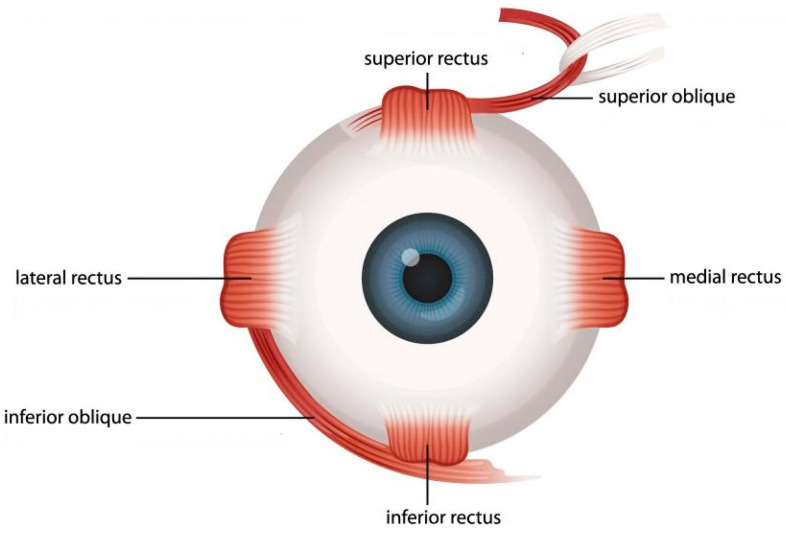
Extraocular muscles.

**Figure 2 ijms-25-04985-f002:**
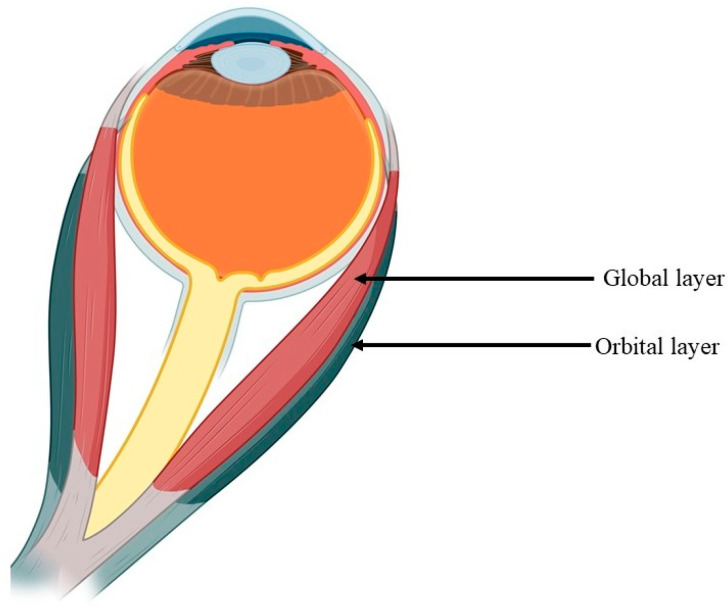
The division of the EOMs into two layers: the “orbital” layer and the “global” layer.

**Figure 3 ijms-25-04985-f003:**
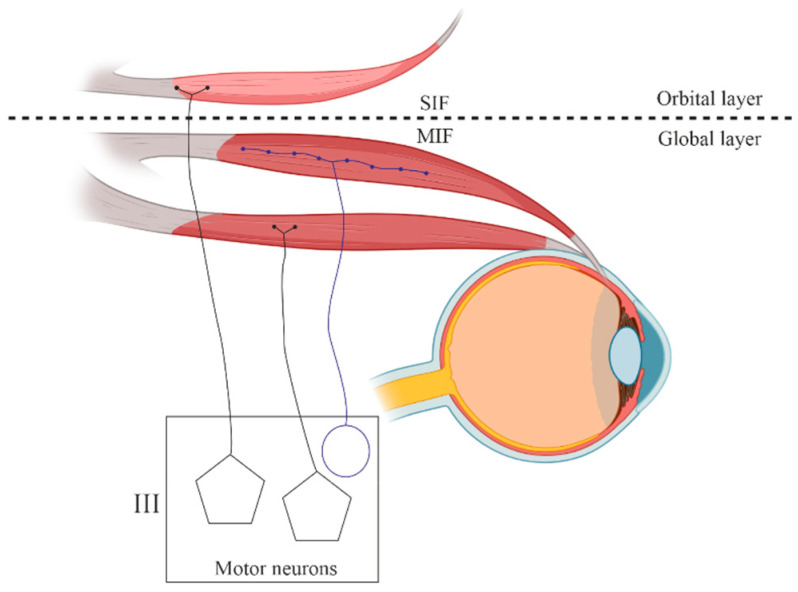
Schematic morphology of the innervation of the extraocular muscles. III—oculomotor nucleus; IIIn, third cranial nerve. SIF—single innervation fiber, MIF—multiple innervation fibers.

**Figure 4 ijms-25-04985-f004:**
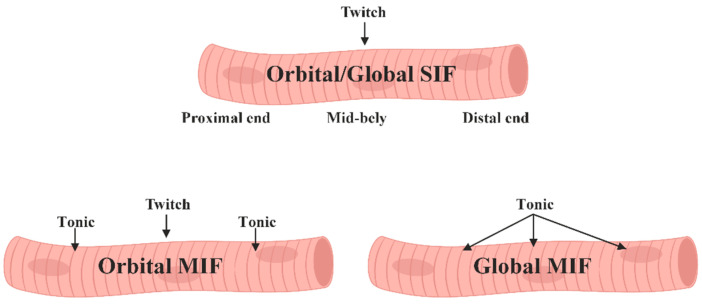
Schematic representation of various extraocular muscle fiber types contraction mechanisms (SIF—singly innervated fiber; MIF—multiply innervated fiber).

**Figure 5 ijms-25-04985-f005:**
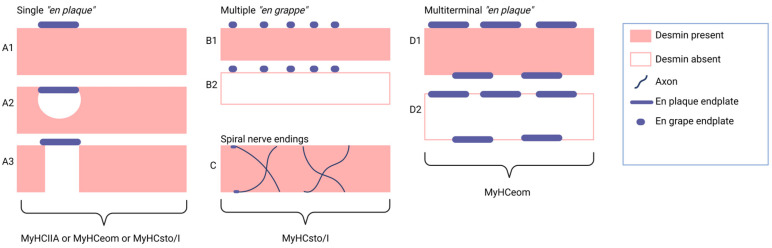
Schematic illustrations depict the desmin composition patterns in three types of human EOM fibers. (**A**) demonstrates (**A1**) uniform desmin staining; (**A2**) absence of desmin in a confined area; or (**A3**) absence in a larger area proximal to the NMJ, with uniform staining in the remaining muscle fiber. (**B**) Desmin was typically present in MyHCsto/I fibers (**B1**) but absent in a subset of these fibers (**B2**). (**C**) Desmin was consistently found in myHCsto/I myofibers with spiral nerve endings. (**D**) Desmin was present (**D1**) or absent (**D2**) in muscle fibers containing MyHCeom.

**Table 1 ijms-25-04985-t001:** MyHC isoforms in EOMs.

Type of Isoform	Name
Fast isoforms of MyHCs	MyHCIIa (*Myh2*); MyHCIIb (*Myh4*); MyHCIIx (*Myh1*); MyHCeom (*Myh13*).
Slow isoforms of MyHCs	MyHCI-MyHC-β/slow (*Myh7*); MyHCα-cardiac (*Myh6*); MyH14/7b (*MyH7B*)
MyHC isoforms related to development	MyHCemb (*Myh3*); MyHCneonatal (*Myh8*)
The ancient sarcomeric MyHCs	MYH14/7b *(MyH7B)*, MyH15 (*Myh15*)

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
