# Peer review of "Extreme Tolerance of Extraocular Muscles to Diseases and Aging: Why and How?"

_ijms, 2024, doi:10.3390/ijms25094985_

Round 1

Reviewer 1 Report

Comments and Suggestions for Authors

Table 1: “MyHC-B” should be changed to “MyHC-beta” (fine to use the Greek symbol). Also, “MyHCsto” should be spelled out somewhere. Presumably this is the slow tonic isoform of MyHC, but this is never stated. The MyHC7B gene is associated with two isoforms – MyHCsto and MyHC7b. Is this correct?

Line 113 to 116: The classification is at the level of fiber types, not at the EOM level. Therefore, in line 113, it should be “a classification of EOM fiber types…”. Also, “(Extraocular Muscles)” is not needed here because “EOM” is already defined in line 37.

Line 126-131: The authors refer to classification of EOM and distinct EOM types but this should be changed to “EOM fiber types”. Also, in line 127, it should be “…only five fiber types in EOM…”.

The authors should define “polyneuronal innervation” and “multiple innervation” which are different from each other and then use the correct term, beginning with line 157.

Figure 3 is not needed as it does not illustrate elimination of polyneuronal innervation.

Line 179: the authors refer to “multitendinous endings. This should be clarified or corrected.

Line 186: “influenced by” should be changed to “associated with”.

Line 255: The authors state that desmin is not present in some human EOM muscle fibers. Is this true for at only neuromuscular junctions or at Z disks, as well?

Line 323” it is not clear to what “MBs”is referring. Should this be “MFs”?

Line373: “are” should be changed to “is”.

The sentence in lines 476-478 is not clear and requires editing. Specifically, “Intriguingly, the process of MF type transition while preserving innervation between MyHCeom and neuronal cells” is not clear.

Line 489” “skeletal” should be inserted after “any”.

Comments on the Quality of English Language

Please see my suggestions for editing in my comments to the authors.

Author Response

Dear Reviewer,

Thank you for your thorough review of our manuscript titled " Extreme tolerance of extraocular muscles to diseases and aging: why and how?". We appreciate the time and effort you have invested in providing valuable feedback to improve the quality of our work.

We have carefully considered each of your suggestions and comments and are committed to addressing them to enhance the clarity and accuracy of our research findings. Below, we provide our responses and planned actions regarding each point raised:

  1. Table 1: We acknowledge the need to change "MyHC-B" to "MyHC-beta" and ensure that "MyHCsto" is spelled out and properly defined as the slow tonic isoform of MyHC, and we actually we changed it to MуH14/7b.
  2. Line 113 to 116: We revised the text to accurately reflect that the classification pertains to fiber types within the EOM and removed the unnecessary abbreviation "(Extraocular Muscles)."
  3. Line 126-131: We modified the text to refer to "EOM fiber types" and corrected the statement regarding the number of fiber types in EOM.
  4. Polyneuronal Innervation: We will define "polyneuronal innervation" and "multiple innervation" as requested, and ensure the correct term is used consistently throughout the manuscript.
  5. Figure 3: We deleted the figure 3.
  6. Multitendinous Endings: We provided correction regarding "multitendinous endings" and changed it to “multiply innervated muscle fibers”
  7. Line 186: We changed "influenced by" to "associated with" in line 186 for accuracy.
  8. Desmin in EOM Muscle Fibers: Based on the information provided, the absence of desmin can indeed be observed in some parts of human extraocular muscle fibers, not only at neuromuscular junctions but also in proximity to Z disks. However, this absence may vary among different muscle fibers. In certain cases, desmin may be completely absent, while in others, its absence may be localized near neuromuscular junctions. The study conducted by Jing-Xia Liu et al. (PMID: 32176266) supports this observation, demonstrating the absence of desmin in various regions of muscle fibers, including both neuromuscular ends and along the length of the fiber, as well as in relation to Z-disks.
  9. Line 323: We corrected it.
  10. Line 373: We corrected it.
  11. Additionally, we carefully edited and rephrased the sentence in lines 476-478 to improve clarity and gave the references to it.
  12. Line 489: We corrected it.

Once again, we thank you for your insightful comments and constructive feedback. We are confident that addressing these issues will strengthen the overall quality of our manuscript.

Sincerely, Dr. A. Bilyalov.

Reviewer 2 Report

Comments and Suggestions for Authors

The authors showed in this review the biochemistry and structural difference between skeletal muscle and EOM. Furthermore, the authors are also demonstrating why EOM has tolerance to disease and aging.

In my opinion, this review is an important compilation of what is available in the literature about the EOM and the important characteristics of this muscle.

I only have 2 points to highlight:

- In item 1.1, I would like to suggest that the authors add to the text the name of 6 muscles responsible for controlling the movement of the eyeball, as it is currently only in the figure.

- Regarding of figure 3, I cannot visualize the elimination of the polyneuronal innervation of skeletal muscles in this figure, I suggest the authors add a figure showing the before and after the elimination.

Author Response

Dear Reviewer,

Thank you for taking the time to review our manuscript titled " Extreme tolerance of extraocular muscles to diseases and aging: why and how?" We appreciate your insightful feedback and constructive suggestions. We have carefully considered your comments and made revisions accordingly. Below, we address each of your points:

  1. Addition of Muscle Names in Section 1.1:

We acknowledge your suggestion regarding the inclusion of the names of the six muscles responsible for controlling the movement of the eyeball in Section 1.1. We agree that this addition will enhance the clarity and completeness of the text. Therefore, we have incorporated the names of these muscles into the relevant section of the manuscript.

  1. Visualization of Polyneuronal Innervation Elimination in Figure 3:

We have deleted this Figure, because the same information is presented in the next Figure.

We believe that these revisions strengthen the manuscript and address the points raised by the reviewer. We are confident that these enhancements will improve the clarity and comprehensiveness of our work.

Once again, we thank you for your valuable feedback and for your contribution to the peer-review process. Should you have any further questions or require additional information, please do not hesitate to contact us.

Sincerely, Dr. A. Bilyalov.